# Remote Sensing Provides a Rapid Epidemiological Context for the Control of African Swine Fever in Germany

**DOI:** 10.3390/s23198202

**Published:** 2023-09-30

**Authors:** Hannes Bergmann, Eva-Maria Czaja, Annett Frick, Ulf Klaaß, Ronny Marquart, Annett Rudovsky, Diana Holland, Patrick Wysocki, Daike Lehnau, Ronald Schröder, Lisa Rogoll, Carola Sauter-Louis, Timo Homeier-Bachmann

**Affiliations:** 1Friedrich-Loeffler-Institut, Federal Research Institute for Animal Health, Institute of Epidemiology, Südufer 10, 17493 Greifswald-Insel Riems, Germany; hannes.bergmann@fli.de (H.B.); eva-czaja@gmx.de (E.-M.C.); patrick.wysocki@fli.de (P.W.); daike.lehnau@gmail.com (D.L.); ronald.schroeder@fli.de (R.S.); lisa.rogoll@fli.de (L.R.); carola.sauter-louis@fli.de (C.S.-L.); 2LUP-Luftbild Umwelt Planung GmbH, Große Weinmeisterstraße 3a, 14469 Potsdam, Germany; annett.frick@lup-umwelt.de; 3Landesamt für Arbeitsschutz, Verbraucherschutz und Gesundheit, Abteilung Verbraucherschutz, Dezernat V2, Dorfstraße 1, 14513 Teltow OT Ruhlsdorf, Germany; ulf.klaass@lavg.brandenburg.de (U.K.); ronny.marquart@lavg.brandenburg.de (R.M.); annett.rudovsky@lavg.brandenburg.de (A.R.); diana.holland@lavg.brandenburg.de (D.H.)

**Keywords:** African swine fever, epidemiology, remote sensing, geographical information systems, risk factor, risk assessment

## Abstract

Transboundary disease control, as for African swine fever (ASF), requires rapid understanding of the locally relevant potential risk factors. Here, we show how satellite remote sensing can be applied to the field of animal disease control by providing an epidemiological context for the implementation of measures against the occurrence of ASF in Germany. We find that remotely sensed observations are of the greatest value at a lower jurisdictional level, particularly in support of wild boar carcass search efforts.

Introduction

African swine fever (ASF) is an internationally spreading, viral pig disease that severely damages agricultural pork production, economy and social welfare [1]. The ASF virus (ASFV) infects pigs, including wild boar, but is not harmful to humans. ASFV infection in pigs results in high virus concentrations in the blood and leads to generalised internal bleeding that usually kills affected pigs within several days [2]. In Europe, wild boars are considered important in the epidemiology of ASFV, from which it is assumed that the disease occasionally transfers into domestic pig holdings [3]. Considering wild boar biology and movement, the transmission of ASFV amongst wild boar could explain the dominating gradual spread of ASF observed in the current European epidemic. The complex behaviour of wild boar, uncertainties around the relevant ASF spread mechanisms and the current lack of a vaccine have made ASF difficult to control [4].

In the current epidemic, ASF has progressively spread across Europe and Asia from the original outbreak in Georgia in 2007. The ASF epidemic front in Europe has travelled from east to west and formed a satellite outbreak in West Poland in November 2019, approaching Germany. This threat to Germany elicited a sophisticated assortment of preventive disease control measures. Nevertheless, on 10 September 2020, near the Polish border, a wild boar carcass tested positive for ASF in Germany, marking the first ASF occurrence in this country [5].

Following the discovery of the disease, pre-planned responses were initiated. One of the key tasks was to determine the extent of the infected area as quickly as possible and based on this information, implement hazard and risk zones in the affected area. 

In the past, mainly retrospective or static data, for example on land use, were applied for setting up these restriction zones (as mentioned in the European Commission working document on principles and criteria for geographically and temporally defining ASF regionalisation—working document SANTE/7112/2015) [6]. However, the currency of these approaches would be variable and largely ignore the important spatio-temporal fluidity of the wild boar habitats to implement effective ASF control measures.

When facing a disease incursion event, such as ASF, it is therefore critical to gain a timely and spatially explicit overview of the affected area. Systematic searches for wild boar carcasses and testing these for the presence of ASFV were, and continue to be, conducted throughout the risk area to guide the process of delimitating zones and to control the disease by removing infectious carcass material. Due to limited resources (e.g., personnel, finances), prioritization of the areas with high attractiveness for wild boars and consequentially increased permeability for ASF spread can increase the efficiency of targeted searches [7]. The attractiveness of the landscape for wild boar depends on various factors, some of which vary seasonally, e.g., maize fields, acorn/beech mast. Rather than using retrospective or static data, as has been used in the past, dynamic up-to-date information provides the opportunity to target the prioritization of such searches explicitly.

Besides sporadic outbreaks of ASF in domestic pigs, mainly wild boar have been affected by ASF in Germany [8,9]. Several carcasses of wild boar that tested positive for ASFV were found in harvested maize fields. Maize fields offer shelter and food for wild boar, and thus are an attractive habitat for this species. Once maize fields are harvested, the wild boar have to leave and find new areas to live in. Thus, for setting up restriction zones and control zones as regulated in the Commission Implementing Regulation (EU) 2023/594, it is of paramount importance to know where maize crops are located at the time and where wild boar could have moved from recently harvested fields. It is conceivable that wild boar will move into other still existing maize fields, or into forests that offer food and shelter, such as forests with oak and beech trees [10,11]. Remotely sensed observations can provide insights into important wild boar habitat factors, including food, water and shelter.

The use of satellite-based remote sensing data is one way to obtain up-to-date information quickly and easily. Conceptually, remote sensing has proven to be an effective tool for monitoring agricultural production. Due to a large variety of on-board sensors on an increasing number of civilian satellites, the spectral and temporal properties of the land surface resulting from human practices can be captured and monitored at different spatial and temporal scales [12]. Remote sensing is commonly applied in the field of agricultural crop production, including the monitoring of crop growth and detection of crop stress [13]. In addition, the application of remote sensing is well established in forest science for forest biomass monitoring [14] or forest tree species composition [15]. There are also approaches to use remote sensing data for early warning systems, e.g., remotely sensed sea water surface temperature as a predictor of the risk of Vibrio infections [16]. Whilst a considerable variety of remote sensing data is available and climate change instils an increasingly pressing need to interpret this information in a veterinary epidemiology context, many barriers still exist that prevent the wider use of such data for emerging disease management [17]. To our knowledge, this is particularly relevant for transferring satellite-based remote sensing technologies beyond research applications to the animal disease control sector in the field.

Here, we present how remotely sensed satellite observations can be applied for regional risk assessment in the context of ASF control in Germany and how remote sensing data are quickly prepared and provisioned to the competent authorities in the ASF outbreak area. A particular goal of this study was to better understand the relevance of satellite-based remote sensing for disease control efforts that followed the ASF incursion into eastern Germany on 10 September 2020.

## 2. Materials and Methods

### 2.1. Crop Classification

For the fastest possible provision of current landcover information, the development of fully automated and standardized methods for processing heterogeneous satellite data for large study areas was necessary. The large amount of data to be analysed requires cloud computing services, which provide the necessary data infrastructure and computing power.

Near real-time, remotely sensed information from the European Union’s earth observation program ‘Copernicus’ (https://www.copernicus.eu/en, accessed on 15 November 2021) was acquired. Multitemporal Sentinel-1 Ground Range Detected (GRD) and Sentinel-2 Level 2A (L2A) scenes were used. They provide satellite-based high temporal, spectral and spatial resolution imagery to derive detailed and current land cover information on demand. These data were utilized for the categorization of primary crop types (refer to Table 2) and for assessing the current status of maize harvesting, as well as for identifying the presence and distribution of oak and beech trees. The primary aim of the classification was to rapidly and precisely predict those crucial landcover characteristics, all while upholding a high degree of spatial accuracy. This information was specifically directed towards regions in Germany where cases of African Swine Fever (ASF) in wild boars had been reported along the German–Polish border. Its purpose was to pinpoint suitable habitats for wild boars in those areas.

The crop type classification Is based on the very effective Random Forest classification algorithm [18]. Each crop type shows different spectral reflection characteristics due to its phenology. The standard characteristics, described by spectral indices, can be used for classification [19,20]. Here, a range of widely used indices were applied in a first model run, including spectral bands and possible band permutations based on the following equation:(1)$$\mathrm{band}\;\mathrm{permutations}=\frac{\left(x-y\right)}{\left(x+y\right)}$$
with *x* and *y* as different spectral bands. All indices tested are listed in Appendix A.

With all these predictors, a Random Forest classifier was trained with 10.000 training points to model the crop type classes (hyperparameters used: 450 trees and minimum leaf population of 4). The permutation-based model’s variable importance, showing the variables with the highest distinctive power, revealed the most useful indices (Table 1). These indices were employed in constructing a conclusive classifier for crop type prediction. This approach resulted in a reduction in the data volume and processing time, leading to an acceleration of the entire workflow.

To evaluate the phenological changes in the index curve throughout the crop cycle, we generated standard curves for all indices in Table 1. The standard curves for the single crop types were derived from multitemporal Sentinel 2 and Sentinel 1 data from 2017 and 2018 by the use of IACS data for several regions in Germany (International Association of Classification Societies—https://iacs.org.uk/, accessed on 15 November 2021). The IACS crop-type classes were aggregated (Table 2). For each class, the corresponding IACS areas were used for the calculation of the statistics (mean, minimum, maximum, standard deviation) for all indices at every date. The mean plus and minus standard deviation values were calculated as well (meanadd, meansub). Through interpolation of the data points and smoothing (2^nd^ polynom), the standard curves “min”, “max”, “mean”, “meanadd” and “meansub” were derived for each index in Table 1 (see NDVI in Figure 1 and Figure 2). Upon examining all the standard curves and their intersections, it became evident that achieving a high classification accuracy would require a dense time series. Since cloudless images are infrequent in northern regions, the aggregation of various scenes became imperative.

The classification was run for aggregated satellite data of two months each, starting on 1 October 2017. In total, a data set of 21 parameters (or 21 layers) was generated for two months each. That data set encompasses the seven spectral indices from Table 1 and all eight VIS, SWIR and NIR spectral bands and the six radar indices. The accuracy of the classification was assessed using the F-score (Table 2). The F-score corresponds to the harmonized mean of precision (rate of true positives vs. all positives) and recall (rate of all true positives vs. true positives and false negatives) and has a value range from 0 (bad) to 1 (perfect). IACS data were used as a reference. 

### 2.2. Preparation and Provisioning of Remote Sensing Data to Competent Authorities

Satellite remote sensing data were prepared to show the location of maize crops, their harvesting status (Figure 3a (20 October 2020), FLI-Maps Links harvesting status: 20 September 2020; 20 October 2020; 12 November 2020) and forest-covered areas as well as their percentage of oak and beech trees on 20 October 2020 (Figure 3b, FLI-Maps Link: oak and beech trees). These data were prepared for the ASF outbreak area at the time with the current ASF control zones and reported wild boar ASF cases considered. The harvest of maize in the ASF outbreak zone was strictly regulated and coordinated to complement ASF control measures. 

Satellite remote sensing allowed spatial tracking of cropped maize areas over time to inform the implementation of ASF control measures (Figure 4).

The processed satellite information was then distributed through a pre-existing mapping tool known as ‘FLI-Maps’. The tool FLI-Maps was developed by the Friedrich-Loeffler-Institut (FLI) to support Germany’s surveillance obligation to record and control reportable animal diseases. In Germany, reportable animal disease events are recorded in a Geographical Information System (GIS) integrated disease reporting system called TSN (“Tierseuchennachrichtensystem” [28]). TSN utilises the FLI-Maps platform to geospatially summarise the status of reportable diseases and is readily accessible to relevant veterinary authorities in Germany.

### 2.3. Evaluating the Relevance of Remote Sensing Data for ASF Control

To evaluate the application of remotely sensed satellite observations in the early control phases following the month after ASF incursion in eastern Germany on 10 September 2020, an electronic questionnaire was administered to jurisdictional key personnel within the veterinary authorities engaged with ASF control in Germany. The questionnaire was circulated in the beginning of March 2021, thus capturing experiences from the first six months of ASF management in the entire affected area of Eastern Germany. The questionnaire is included with this article as Appendix A (Appendix A). It elicited the relevance of utilising remote sensing data by scoring different ASF management applications during the current outbreak management, as well as the jurisdictional working level of the respondent. The respondents were able to select semi-quantitative responses (rank 0, 1, 2, 3; according to the four response options in the questionnaire, see Appendix A) regarding their experiences of applying satellite-based remote sensing data to ASF management, and to what extend this technology influenced their management of the disease. The relevance of remote sensing data for each queried ASF management-related application was calculated by summing the cumulative scores provided from all the respondents and presenting it as a proportion of the possible maximum score, stratified by jurisdictional level. The maximum score is given by the number of respondents multiplied with the highest possible rank 3.

## 3. Results

### 3.1. Accuracy of Crop Classification

When looking at the F-scores of the individual crop types (Table 2), it was noticeable that the quality of the classification of certain crop types also depended on the amount of satellite data used. Maize, rye, barley and sugar beet were identified very reliably if data from more than 8 months were available. Grassland, wheat and rapeseed, on the other hand, were correctly classified with less data. In the case of oats, fallow land and potatoes, only mediocre results were achieved at the end of the study period (30 September). The highly heterogeneous and mixed classes that grouped other crop types together hardly achieved any usable accuracies.

### 3.2. Evaluation of Relevance of Remote Sensing Data

At the lower jurisdictional level (district), the value of applying satellite remote sensing was assessed to be very relevant for the selection of areas targeted during the wild boar carcass searches for the choice of carcass search method and to guide the positioning of wild boar fencing (Figure 5). Overall, the value of remote sensing application in this context appeared to be of the greatest relevance in the field at lower jurisdictional levels, whereas representatives of higher-level jurisdictional authorities reported lower relevance of the technology for ASF control by comparison (Figure 5).

## 4. Discussion

Infectious disease epidemics result from direct interaction with environmental factors through space and time. It is therefore necessary to map the epidemiological context of environmental factors to have the best chance of comprehending relevant disease patterns suitable for intervention during disease incursion events. Whilst remotely sensed and disease-risk-related geospatial information is usually available to authorities, these types of data tend to be used extensively in retrospect only, rather than during an acute outbreak phase. 

Applying and regularly updating (Figure 3 and Figure 4) remotely sensed satellite data in several applications was found to be relevant for the implementation of ASF control measures in Germany, particularly at lower jurisdictional levels (Figure 5). Remote satellite sensing was applied to map the current extent of maize crops and the distribution of oak and beech forest in the ASF outbreak area. The prepared information was readily shared with the responsible authorities through the pre-existing FLI-Maps tool that had been integrated into the governmental veterinary TSN management software (TSN 3.3 R7a). In the field, remote sensing was applied to the selection of wild boar search areas and methods, including the planning and targeting of search missions by drones, helicopters, sniffer dogs and trapping teams. Remote sensing was also relevant for the positioning of wild boar fencing and monitoring compliance with the maize harvest ban regulations. As such, satellite-derived remote sensing data offered detailed information to implement risk-based targeting of previously described environmental ASF risk factors for efficient and sustainable disease control efforts [29].

In conclusion, we found that ensuring adequate synthesis and transfer of remotely sensed satellite observations provided a relevant and immediate epidemiological context for acute disease occurrence responses to ASF in the field. Rapid utilisation of GIS and remote sensing systems during the early phase following disease occurrence has the potential to greatly reduce long-term negative effects of such events by appropriately setting the course of disease management early on, likely yielding benefits in disease scenarios other than ASF. We therefore advance a concept to make sharable cartographic platforms and readily available, remotely sensed land cover information an integral part of epidemic preparedness strategies.

## Figures and Tables

**Figure 1 sensors-23-08202-f001:**
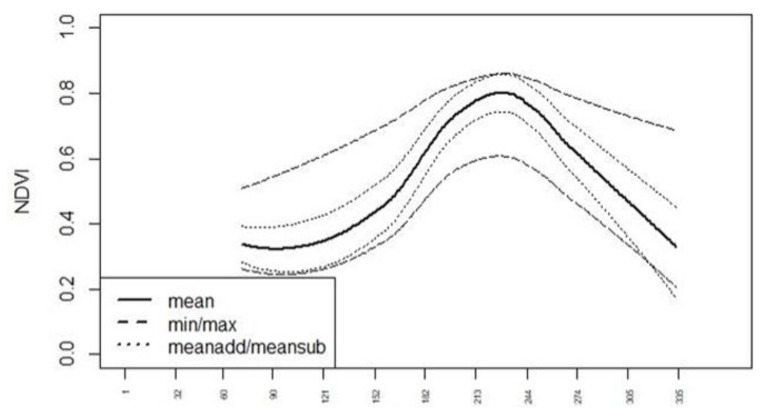
Normalized difference vegetation index (NDVI) standard curves for “corn”, *x*-axis: Day of Year (DOY).

**Figure 2 sensors-23-08202-f002:**
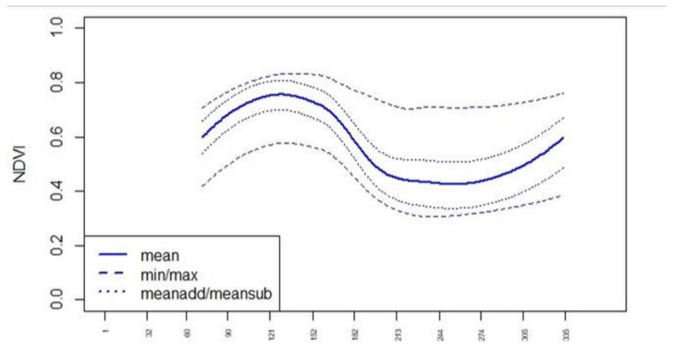
Normalized difference vegetation index (NDVI) standard curves for “canola”. *X*-axis: Day of Year (DOY).

**Figure 3 sensors-23-08202-f003:**
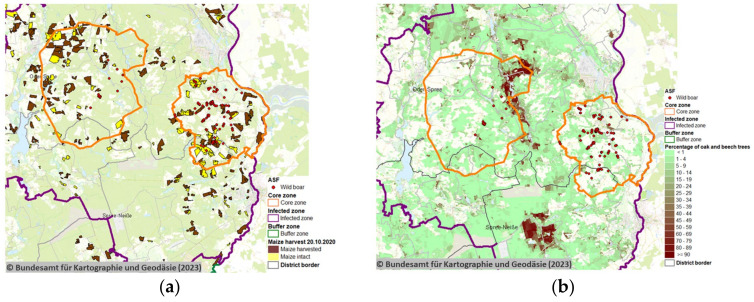
Remote sensing of geospatial disease risk information using satellite-based imagery and geospatial interpretation for the context of the disease event at hand. (**a**) Maize crop harvest status and (**b**) the forest status on 20 October 2020. © Bundesamt für Kartographie und Geodäsie (2023), Datenquellen: https://sgx.geodatenzentrum.de/web_public/Datenquellen_TopPlus_Open (accessed on 7 July 2023).

**Figure 4 sensors-23-08202-f004:**
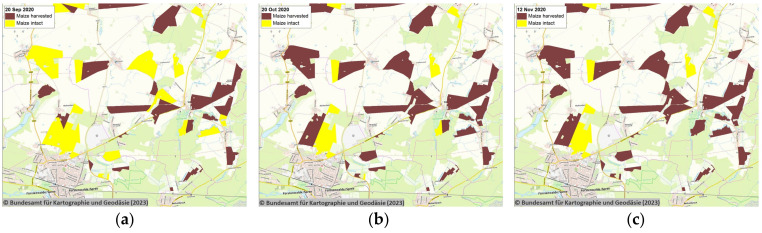
Remote sensing of the ongoing maize crop harvest status on (**a**) 20 September, (**b**) 20 October, and (**c**) 12 November 2020. © Bundesamt für Kartographie und Geodäsie (2023), Datenquellen: https://sgx.geodatenzentrum.de/web_public/Datenquellen_TopPlus_Open (accessed on 7 July 2023).

**Figure 5 sensors-23-08202-f005:**
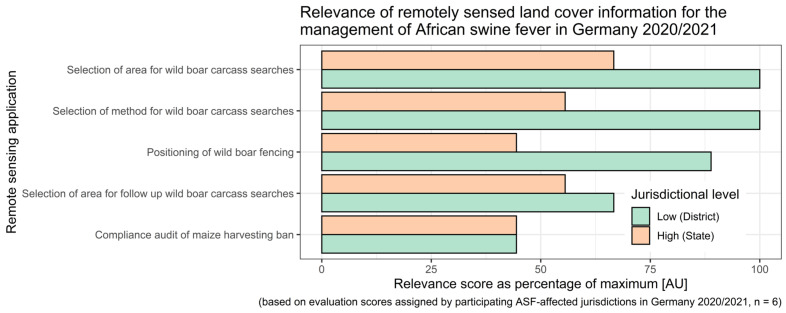
Relevance of remotely sensed land cover information for the management of African swine fever in Germany during 2020/2021. Relevance scores represent a percentage of the maximum score.

**Table 1 sensors-23-08202-t001:** Most useful indices.

Indices	
NDVI	Normalized Difference Vegetation Index [21]
NDYI	Normalized Difference Yellowness Index [22]
GNDVI	Green Normalized Difference Vegetation Index [23]
PVR	Photosynthetic Vigour Ratio [24]
MSAVI	Modified Soil Adjusted Vegetation Index [25]
MSR	Modified Simple Ratio [26]
REIP	Red-Edge Inflection Point [27]
VIS	Band 2, 3, 4
NIR and SWIR	Band 5, 6, 7, 8 and 12
Radar	VV max, VH max, VV/VH-Ratio max, VV Median, VH Median, VV/VH-Ratio Median

**Table 2 sensors-23-08202-t002:** F-scores of the individual crop types for the year 2018 using satellite data from 12, 8 and 4 months, respectively (value range from 0 (bad) to 1 (perfect)).

Crop Type	12 Months	8 Months	4 Months
1: Grassland	0.83	0.76	0.71
2: Fallow land	0.46	0.28	0.22
3: Maize crop	0.87	0.39	0.21
4: Rye	0.74	0.29	0.20
5: Wheat	0.77	0.72	0.50
6: Potato	0.42	0.27	0.14
7: Sugar beet	0.71	0.32	0.24
8: Rapeseed	0.96	0.83	0.58
9: Barley	0.72	0.56	0.38
10: Oats	0.37	0.18	0.06
11: Woody plants	0.21	0.25	0.21
12: Other cereals	0.08	0.04	0.02
13: Root crops, rest	0.11	0.49	0.17

## Data Availability

Publicly available datasets were analyzed in this study. This data can be found here: https://www.copernicus.eu/en; https://iacs.org.uk/. Data concerning the ASF outbreaks are available on request from the corresponding author. The data are not publicly available due to privacy reasons.

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
