# Peer review of "Remote Sensing Provides a Rapid Epidemiological Context for the Control of African Swine Fever in Germany"

_sensors, 2023, doi:10.3390/s23198202_

Round 1

Reviewer 1 Report

Dear authors,

I have had the opportunity to review your manuscript and would like to offer some constructive feedback and suggestions for improvement. Please know that my intention is to help enhance the clarity and impact of your work. You can find my comments in the attached file.

Author Response

Reviewer 1

Dear authors,

I have had the opportunity to review your manuscript and would like to offer some constructive feedback and suggestions for improvement. Please know that my intention is to help enhance the clarity and impact of your work. You can find my comments in the attached file.

We much appreciate the reviewer’s positive and constructive comments, which we felt contributed to enhance the description of the underlying work and will certainly help enhance comprehension of our manuscript for the readers of Sensors.

  1. Objectives of the Manusscript:
  • While it is worth noting that although the benefits of satellite-based remote sensing may not have yet been fully integrated into the animal disease control sector, it has found considerable application in research related to various animal diseases. You wish to highlight this point and provide specific examples from the existing literature to illustrate its potential impact
  • Please consider revising the introduction to clearly state the manuscript’s objectives. Specifically, emphasize the aim to evaluate remote-sensed land cover applications at two administrative levels: state (high) and district (low)

We agree with the reviewer that satellite-based remote sensing does play a considerable role in the animal disease research arena, but to our knowledge, this technology has not been described for the timely management of wild boar and ASF in the current European theatre of events. As such, our paper offers insights into the translation of remote satellite sensing for transboundary animal disease control and the perceived applicability of this technology for the veterinary authorities managing an ASF incursion. Even for current and due to climate change increasingly pressing research applications of remotely sensed environmental data in veterinary epidemiology, many barriers still exist that prevent wider use of such data for disease management. We thus value the reviewer’s insightful remark and include the following text in the revised paper:

“Whilst a considerable variety of remote sensing data is available and climate change instils an increasingly pressing need to interpret this information in a veterinary epidemiology context, many barriers still exist that prevent wider use of such data for emerging disease management [17]. To our knowledge, this is particularly relevant for transferring satellite-based remote sensing technologies beyond research applications to the animal disease control sector in the field”

We further implement the reviewer’s recommendation to clarify the manuscript objective by explicitly describing the aim to evaluate remote-sensing application at two administrative levels. Accordingly, we amend the following text of the introduction:

“Here we present how remotely sensed satellite observations can be applied for regional risk assessment in the context of ASF control in Germany and how remote sensing data is quickly prepared and provisioned to the competent authorities in the ASF outbreak area. A particular goal of this study was to better understand the relevance of satellite-based remote sensing for disease control efforts that followed the ASF incursion into eastern Germany on 10 September 2020.”

And the following sentence in the methods section:

“It elicited the relevance of utilising remote sensing data by scoring different ASF management applications during the current outbreak management, as well as the jurisdictional working level of the respondent.”

  1. Crop Type Classification Methodology
    • Clarify the crop type classification methodology. Mention explicitly whether it was used solely for maize harvest and oak and beech trees or if it was applied to other crops as well. The results section suggests broader application, so this point needs greater clarity.
    • Specify the focus of the classification, whether it is predictions of the most useful indices.
    • In line 96, could you provide a reference for the previous index ranking and clarify which indices were used?
    • In line 123, specify which 21 parameters you are referring to.
    • Consider the placement of Table 2, as its locations in both the Methods and Results sections can be confusing. Streamline the presentation of material in the sections.

We appreciate these differentiated suggestions by the reviewer to clarify the description of our methods. We changed the following text to clarify the focus:

“This data was utilized for the categorization of primary crop types (refer to Table 2) and for assessing the current status of maize harvesting, as well as for identifying the presence and distribution of oak and beech trees. The primary aim of the classification was to rapidly and precisely predict those crucial landcover characteristics, all while upholding a high degree of spatial accuracy. This information was specifically directed towards regions in Germany where cases of African Swine Fever (ASF) in wild boars had been reported along the German-Polish border. Its purpose was to pinpoint suitable habitats for wild boars in those areas.”

We changed the following text to clarify the previous index ranking, which served as a first model run to find the most important indices to minimise data volume and processing time in the final model run:

“Here a range of widely used indices were applied in a first model run, including spectral bands and possible band permutations based on the equation: with x and y as different spectral bands. All indices tested are listed in Supplementary Table 1.”

We added references for all indices used in Table 1 and also clarified their specific use in an additional text:

“These indices were employed in constructing a conclusive classifier for crop type prediction. This approach resulted in a reduction in data volume and processing time, leading to an acceleration of the entire workflow.

To evaluate the phenological changes in the index curve throughout the crop cycle, we generated standard curves for all indices in Table 1. The standard curves for the single crop types were derived from multitemporal Sentinel 2 and Sentinel 1 data from 2017 and 2018 by use of IACS-data for several regions in Germany (International Association of Classification Societies - https://iacs.org.uk/). The IACS crop type classes were aggregated (Table 2). For each class, the corresponding IACS areas served for the calculation of the statistics (mean, minimum, maximum, standard deviation) for all indices at every date. The mean plus and minus standard deviation were calculated as well (meanadd, meansub). Through in-terpolation of the data points and smoothing (2nd polynom), the standard curves “min”, “max”, “mean”, “meanadd” and “meansub” were derived for each index in Table 1 (see exemplarily NDVI in Figure 1 and Figure 2). Upon examining all the standard curves and their intersections, it became evident that achieving a high classification accuracy would require a dense time series. Since cloudless images are infrequent in northern regions, the aggregation of various scenes became imperative.”

We further added clarification on the 21 parameters used:

“The classification was run for aggregated satellite data of two months each, starting on 1 October 2017. In total, a data set of 21 parameters (or 21 layers) was generated for two months each. That data set encompasses the seven spectral indices from table 1 and all eight VIS, SWIR and NIR spectral bands and the six Radar indices. The accuracy of the classification was assessed using the F-score (Table 2).”

  1. Evaluation of Remote Sensing Data Relevance:
  • Consider providing the questionnaire as supplementary material to enhance the transparency of your evaluation.
  • Provide more details about the scoring system used (qualitative, quantitative, and values).
  • Explain why ASF management applications were specifically chosen for maize, oak, and beech trees, and whether this approach applies to other crops.

We thank the reviewer for these suggestions and to include our questionnaire as supplementary material with the paper. We gladly rectify this oversight and provide this information in the revised manuscript. Since the original questionnaire was administered to German authorities, it was written in German language. Here we present the German original wording alongside an English translation as new supplementary information.

The respondents were able to select semi-quantitative responses regarding their experiences of applying satellite-based remote sensing data to ASF management, and to what extend this technology influenced their management of the disease. We include the following text in the methods section to clarify the response options in the questionnaire for the reader:

“The respondents were able to select semi-quantitative responses (rank 0, 1, 2, 3; according to the four response options in the questionnaire) regarding their experiences of applying satellite-based remote sensing data to ASF management, and to what extend this technology influenced their management of the disease. The relevance of remote sensing data for each queried ASF management related application was calculated by summing the cumulative scores provided from all respondents and presenting it as a proportion of the possible maximum score, stratified by jurisdictional level. The maximum score is given by the number of respondents multiplied with the highest possible rank 3.”

The questions about the application of the examined remote sensing data were specifically chosen to focus on known major sources of food for wild boar in these agriculturally managed landscapes. In large areas of the regions and for much time of the year, maize is grown as a common agricultural crop that provides extensive shelter and feed for wild boar, particularly during summer month and until harvest. Thereafter, broad leafed forest environments with oak and beech trees offer attractive refuge and feed for wild boar and as such, oak, beech and maize crop distributions over time are considered to offer relevant drivers of wild boar distribution and hence spread of ASF. It is possible that other crops that could be differentiated through satellite-based remote sensing may also influence the spatio-temporal distribution of wild boar and ASF spread, but only maize, oak and beech distributions as some of the main disease distribution drivers were considered for the work presented in this paper. In line with the following reviewer’s comment, we now expand on this aspect by including the following paragraph to the introduction:

“Besides sporadic outbreaks of ASF in domestic pigs, mainly wild boar have been affected by ASF in Germany [8,9]. Several carcasses of wild boar that tested positive for ASFV were found in harvested maize fields. Maize fields offer shelter and food for wild boar and thus are an attractive habitat for this species. Once maize fields are har-vested, the wild boar have to leave and find new areas to live in. Thus, for setting up re-striction zones and control zones as regulated in the Commission Implementing Regulation (EU) 2023/594, it is of paramount importance to know, where maize crops are located at the time and where wild boar could have moved from recently harvested fields. It is conceivable, that wild boar will move into other still existing maize fields, or into forests that offer food and shelter, such as forest with oak and beech trees [10,11]. Remotely sensed observations can provide insights into important wild boar habitat factors, including food, water and shelter.”

  1. Study Duration and Area:
  • Include the duration of the study period and the extend of the study area explicitly in the Methods and Materials section.

We thank the reviewer for this suggestion. Our study focussed on the ASF situation in Eastern Germany at the beginning of the local epidemic following ASF incursion into the country on 10 September 2020 and the utilisation of satellite remote sensing data for early responses in all the areas that were affected along the German-Polish border. The questionnaire was administered in March 2021, and thus captured experiences made by the competent authorities during the first 5-6 months after the incursion. We update the relevant methods section describing these study details as advised by the reviewer:

“To evaluate the application of remotely sensed satellite observations in the early control phases following the month after ASF incursion in eastern Germany on 10 September 2020, an electronic questionnaire was administered to jurisdictional key personnel within the veterinary authorities engaged with ASF control in Germany. The questionnaire was circulated in the beginning of March 2021, thus capturing experiences from the first six month of ASF management in the entire affected area of Eastern Germany.”

  1. Clarification of Results and Discussion:
  • Some content in the Results section should be better explained in the Methods and Materials section. Pay attention to the paragraph in lines 158-166
  • Similarly, the paragraph in lines 192-203 of the Discussion section would be more suitable as an introduction to your findings.

We welcome these remarks by the reviewer and agree to move the indicated paragraphs to more appropriate locations in the flow of the manuscript. We did this as proposed by the reviewer. The following paragraph (including accompanying figures) has now been adapted and moved to the Materials and Methods section:

“2.2. Preparation and provisioning of remote sensing data to competent authorities

Satellite remote sensing data were prepared to show the location of maize crops, their harvesting status (Figure 3 A (20 October 2020), FLI-Maps Links harvesting status: 20 September 2020; 20 October 2020; 12 November 2020) and forest covered areas as well as their percentage of oak and beech trees on 20 October 2020 (Figure 3 B, FLI-Maps Link: oak and beech trees). These data were prepared for the ASF outbreak area at the time with current ASF control zones and reported wild boar ASF cases considered. The harvest of maize in the ASF outbreak zone was strictly regulated and coordinated to complement ASF control measures. Satellite remote sensing allowed spatial tracking of cropped maize areas over time to inform the implementation of ASF control measures (Figure 4). The processed satellite information was then distributed through a pre-existing mapping tool known as ‘FLI-Maps’. FLI-Maps was developed by the Friedrich-Loeffler-Institut (FLI) to support Germany’s surveillance obligation on recording and controlling reportable animal diseases. In Germany, reportable animal disease events are recorded in a Geographical Information System (GIS) integrated disease re-porting system called TSN (“Tierseuchennachrichtensystem” [28]). TSN utilises the FLI-Maps platform to geospatially summarise the status of reportable diseases and is readily accessible to relevant veterinary authorities in Germany.”

As mentioned and helping with addressing a previous comment, this paragraph has been amended for the introduction as follows:

“Besides sporadic outbreaks of ASF in domestic pigs, mainly wild boar have been affected by ASF in Germany [8,9]. Several carcasses of wild boar that tested positive for ASFV were found in harvested maize fields. Maize fields offer shelter and food for wild boar and thus are an attractive habitat for this species. Once maize fields are har-vested, the wild boar have to leave and find new areas to live in. Thus, for setting up re-striction zones and control zones as regulated in the Commission Implementing Regulation (EU) 2023/594, it is of paramount importance to know, where maize crops are located at the time and where wild boar could have moved from recently harvested fields. It is conceivable, that wild boar will move into other still existing maize fields, or into forests that offer food and shelter, such as forest with oak and beech trees [10,11]. Re-motely sensed observations can provide insights into important wild boar habitat factors, including food, water and shelter.”

  1. Paper Strength and Depth
  • While your paper’s main objective is to demonstrate the utility of real-time remote sensor global data in animal health management decisions, consider providing a deeper analysis. This could involve statistical comparisons or epidemiological inferences regarding areas with and without the tested environmental layer.

We thank the reviewer for this interesting suggestion. It would be very to revealing to compare the effects of ASF control measures in areas to which satellite-based remote sensing has been utilised with areas in which this has not been done. Although such a study would be very helpful to understand the contributions of remote-sensing data, we feel that it would be difficult to establish such a comparison in eastern Germany and even if this would be possible, that in retrospect this type of analysis would exceed the scope of our paper. To our knowledge, some type of remote-sensing data has been used in most ASF control situations in Germany, which would make it difficult to identify differentially treated outbreak areas for analysis. Additionally, it is considered challenging to measure the success and efficiency of ASF control efforts, which would be needed to detect any differences in response to application of remote-sensing data. In a similar vein, we understand that the reviewer’s proposal also hints at possibly identifying environmental risk factors for the occurrence of ASF in wild boar populations. We agree that environmental disease risk factors are critical for efficient and sustainable ASF control efforts in this context and that satellite remote-sensing could provide accurate spatio-temporal information regarding such risk factors and thus include the following text to highlight this role of for remote-sensing data application:

“As such, satellite derived remote sensing data offered detailed information to implement risk-based targeting of previously described environmental ASF risk factors for efficient and sustainable disease control efforts.”

In conclusion, I believe your work has potential, and with some revisions to address the above points, it can greatly enhance its clarity and impact. Additionally, you might consider the paper would be better suited for a short communication format given its focus and objectives.

We thank the reviewer again for the encouraging and helpful comments and hope that we were able to address all of them satisfactorily for the audience of Sensors. In line with the reviewer’s suggestion, we have submitted our manuscript as a short communication, rather than a full article.

Reviewer 2 Report

Remote sensing technology is potential in the control of ASF, which is a valuable and important task. The authors extracts crop spatial distribution with Sentinel-2 remote sensing images and relate it with ASF spreading. I have some concerns below need to be addressed.

1.     For the Introduction part, the content of ASF occupies most. While the novelty of this paper lies in the usage of remote sensing images to address the problem of disease spreading, so the latest remote sensing based works on disease supervision should be added.

2.     In Line 93, Page 2, the crop type classification procedure was done by RF classifier, this kind of classifier has many hyper-parameters to be set. The authors should provide them in a table or in other ways.

3.     In Line 104, Page 3, many indices are not referenced except for NDVI and NDYI, which could be added in the table caption. Except that, how many indices actually are used in the RF model training procedure and classification? In Figure 1 and Figure 2, only NDVI statistical curves are presented, what is the role of other indices?

4.     The quality of Figure 5 needs to be improved, and it is hard to distinguish the texts in the figure. The discussion towards crop classification results and ASF is quite weak, only Figure 5 is not enough to support the conclusion, more quantitative experiments is needed.

English quality does not affect the comprehension of the manuscript.

Author Response

Remote sensing technology is potential in the control of ASF, which is a valuable and important task. The authors extracts crop spatial distribution with Sentinel-2 remote sensing images and relate it with ASF spreading. I have some concerns below need to be addressed.

  1. For the Introduction part, the content of ASF occupies most. While the novelty of this paper lies in the usage of remote sensing images to address the problem of disease spreading, so the latest remote sensing based works on disease supervision should be added.

We thank the reviewer for this suggestion, which is in line with a comment of the other reviewer. Accordingly, we have included additional text in the relevant section to provide a link and perspective to the current state of play with regards to satellite remote sensing technology and its applicability to veterinary epidemiology and transboundary animal disease management:

“Whilst a considerable variety of remote sensing data is available and climate change instils an increasingly pressing need to interpret this information in a veterinary epidemiology context, many barriers still exist that prevent wider use of such data for emerging disease management [17]. To our knowledge, this is particularly relevant for transferring satellite-based remote sensing technologies beyond research applications to the animal disease control sector in the field.”

  1. In Line 93, Page 2, the crop type classification procedure was done by RF classifier, this kind of classifier has many hyper-parameters to be set. The authors should provide them in a table or in other ways.

We thank the reviewer for this suggestion and added the following clarification:

“With all these predictors a random forest classifier was trained with 10.000 training points to model the crop type classes (hyperparameters used: 450 trees and minimum leaf population 4).”

  1. In Line 104, Page 3, many indices are not referenced except for NDVI and NDYI, which could be added in the table caption. Except that, how many indices actually are used in the RF model training procedure and classification? In Figure 1 and Figure 2, only NDVI statistical curves are presented, what is the role of other indices?

We appreciate and agree with the reviewer’s remarks. We added references for all indices used in Table 1 and also clarified their specific use in an additional text:

“These indices were employed in constructing a conclusive classifier for crop type prediction. This approach resulted in a reduction in data volume and processing time, leading to an acceleration of the entire workflow.

To evaluate the phenological changes in the index curve throughout the crop cycle, we generated standard curves for all indices in Table 1. The standard curves for the single crop types were derived from multitemporal Sentinel 2 and Sentinel 1 data from 2017 and 2018 by use of IACS-data for several regions in Germany (International Association of Classification Societies - https://iacs.org.uk/). The IACS crop type classes were aggregated (Table 2). For each class, the corresponding IACS areas served for the calculation of the statistics (mean, minimum, maximum, standard deviation) for all indices at every date. The mean plus and minus standard deviation were calculated as well (meanadd, meansub). Through in-terpolation of the data points and smoothing (2nd polynom), the standard curves “min”, “max”, “mean”, “meanadd” and “meansub” were derived for each index in Table 1 (see exemplarily NDVI in Figure 1 and Figure 2). Upon examining all the standard curves and their their intersections, it became evident that achieving a high classification accuracy would require a dense time series. Since cloudless images are infrequent in northern regions, the aggregation of various scenes became imperative.”

  1. The quality of Figure 5 needs to be improved, and it is hard to distinguish the texts in the figure. The discussion towards crop classification results and ASF is quite weak, only Figure 5 is not enough to support the conclusion, more quantitative experiments is needed.

We appreciate and agree with the reviewer’s remarks regarding the image quality of Figure 5. We therefore paste a higher resolution image of this figure into the revised manuscript. We will be eager to provide a suitable image quality as deemed necessary by the journal during the final stages of manuscript processing before publication. It will be easily possible to tailor image quality to the needs of the journal.

We further understand the reviewer’s desire for additional data regarding the role and influence of remote-sensing data on ASF control in eastern Germany. This comment is also in line with a suggestion made by the other reviewer of this paper. We fully agree, that quantitative analysis of the effect remote sensing data utilisation may have on ASF control and disease spread. However, given the fact that these types of data were supplied for the entire study area, thus depriving us of a suitable comparator, and the difficulty of measuring/quantifying efficacy of ASF control we hope the reviewer will be able to accept our indeed limited approach to evaluating the role of satellite remote sensing in this context. Within the given scope of this short communication, additional value stems from describing the novelty of transferring remote sensing technology to ASF control in the field.

Round 2

Reviewer 1 Report

Thank you for considering my feedback and for your cooperation in addressing my suggestions and incorporating them into the revised manuscript. Your responsiveness to the feedback has contributed significantly to the manuscript's improvement. I am satisfied with the revisions made, and it is my pleasure to recommend the manuscript for publication in Sensors. 

Reviewer 2 Report

The authors have resolved my concerns.